# Riemannian Geometry of Multimodal Biometric Embedding Spaces

**Alok Upadhyay**
Birla Institute of Technology & Science – Pilani
f2009583g@alumni.bits-pilani.ac.in

## Abstract

Pretrained biometric encoders project faces and voices into high-dimensional Euclidean spaces, yet their outputs concentrate near low-dimensional Riemannian submanifolds whose geometry is poorly understood. We characterize the intrinsic geometry of seven face and voice encoders—measuring intrinsic dimensionality, local curvature via the second fundamental form, and cluster topology—and ask whether these geometric quantities predict cross-modal person-matching difficulty. Across 12 encoder pairs on VoxCeleb1 (1,249 identities), CKA similarity correlates with cross-modal equal error rate (EER) at Spearman $\rho = -0.87$ ($p < 0.001$), and a multivariate model achieves leave-one-out cross-validated $R^2 = 0.77$. These results suggest that intrinsic geometry provides an informative, *task-agnostic* predictor of cross-modal matching difficulty.

## 1 Introduction

Modern biometric systems rely on deep encoders that map raw signals—face images and speech utterances—into fixed-dimensional Euclidean embedding spaces $\mathbb{R}^d$. Matching is then reduced to nearest-neighbor search under the cosine metric. Although these systems achieve remarkable accuracy within a single modality, cross-modal person matching (e.g., linking a face to a voice) remains challenging: the two embedding spaces encode identity through fundamentally different physical processes, and there is no guarantee that their geometric structures are compatible.

A growing body of work has shown that pretrained embeddings do not fill their ambient space uniformly; instead, they concentrate near low-dimensional, often curved, submanifolds (Pope et al., 2021; Ansuini et al., 2019). Understanding the *intrinsic Riemannian geometry* of these submanifolds—their dimensionality, curvature, and topology—may therefore reveal why some encoder pairs lend themselves to cross-modal alignment while others do not.

**Contributions.**

1. We estimate the intrinsic dimensionality, local curvature (mean and Gaussian), and cluster topology of seven pretrained face and voice encoders on VoxCeleb1.
2. We define and compute four cross-modal geometric divergences: Gromov–Wasserstein distance, spectral gap divergence, CKA, and intrinsic-dimensionality mismatch.
3. We establish that these geometric quantities predict cross-modal EER (via CCA-aligned cosine similarity) with Spearman $\rho = -0.87$ ($p < 0.001$), suggesting a path toward geometry-aware encoder selection and design.

## 2 Related Work

**Intrinsic dimensionality estimation.** The intrinsic dimensionality (ID) of data manifolds has been studied extensively. Levina & Bickel (2004) proposed a maximum-likelihood estimator (MLE) based on $k$-nearest-neighbor distances. Facco et al. (2017) introduced the TwoNN estimator, which exploits the Pareto distribution of the ratio of the two nearest-neighbor distances and is robust to density variations. Recent work has applied these estimators to deep-learning representations (Ansuini et al., 2019; Pope et al., 2021), revealing that ID often varies across layers and training stages.

**Manifold learning and embedding geometry.** The geometric properties of learned representations have attracted increasing attention. Bronstein et al. (2017) provide a comprehensive survey of geometric deep learning. Recanatesi et al. (2019) study how representation dimensionality relates to task performance. Local curvature has been estimated for point clouds in computational geometry (Cazals & Pouget, 2005) and more recently for neural-network loss landscapes (Li et al., 2018).

**Cross-modal biometrics.** Face–voice association has been studied as a cross-modal retrieval problem (Nagrani et al., 2018b;a; Kim et al., 2021). These works typically learn joint embedding spaces via supervision; our work instead studies the intrinsic geometry of *pretrained, independently trained* encoders without additional cross-modal supervision.

**Riemannian geometry in machine learning.** Riemannian methods appear in metric learning (Harandi et al., 2018), generative modeling (Chen et al., 2018), and optimal transport (Mémoli, 2011; Peyré & Cuturi, 2019). Our work is closest in spirit to Chen et al. (2018), who analyze the Riemannian structure of generative-model latent spaces, but differs in that we study discriminative biometric embeddings and focus on cross-modal predictability.

## 3 METHODS

### 3.1 NOTATION AND SETUP

Let $\phi_f : \mathcal{X}_f \to \mathbb{R}^{d_f}$ and $\phi_v : \mathcal{X}_v \to \mathbb{R}^{d_v}$ denote pretrained face and voice encoders, respectively. Given a dataset of $N$ identities, each with paired face images and voice segments, define the empirical embedding sets

$$X_f = \{\boldsymbol{x}_i^f = \phi_f(\text{face}_i)\}_{i=1}^n, \qquad X_v = \{\boldsymbol{x}_i^v = \phi_v(\text{voice}_i)\}_{i=1}^n. \tag{1}$$

We model the support of each empirical distribution as a compact Riemannian submanifold

$$\mathcal{M}_f \hookrightarrow \mathbb{R}^{d_f}, \qquad \mathcal{M}_v \hookrightarrow \mathbb{R}^{d_v}, \tag{2}$$

equipped with the metric $g$ induced by the Euclidean inner product on the ambient space. Concretely, for $p \in \mathcal{M}_f$ and tangent vectors $u, v \in T_p \mathcal{M}_f$,

$$g_p(u, v) = \langle u,\ v \rangle_{\mathbb{R}^{d_f}}. \tag{3}$$

In practice, all embeddings are $\ell_2$-normalized, so the submanifolds lie on the unit sphere $\mathbb{S}^{d-1}$; the analysis below applies to the extrinsic geometry of $\mathcal{M} \hookrightarrow \mathbb{R}^d$ (equivalently, the intrinsic geometry of $\mathcal{M}$ with the round metric restricted from $\mathbb{S}^{d-1}$).

### 3.2 INTRINSIC DIMENSIONALITY ESTIMATION

**Definition 1** (MLE estimator (Levina & Bickel, 2004)). *Let $r_1(x) \leq r_2(x) \leq \cdots \leq r_k(x)$ denote the distances from point $x$ to its $k$ nearest neighbors. The MLE intrinsic-dimensionality estimator at $x$ is*

$$\hat{m}_{\text{MLE}}(x) = \left( \frac{1}{k-1} \sum_{j=1}^{k-1} \log \frac{r_k(x)}{r_j(x)} \right)^{-1}. \tag{4}$$

*The global estimate is obtained by averaging over a sample: $\hat{m}_{\text{MLE}} = \frac{1}{n} \sum_{i=1}^n \hat{m}_{\text{MLE}}(x_i)$.*

**Definition 2** (TwoNN estimator (Facco et al., 2017)). *Define the distance ratio $\mu(x) = r_2(x)/r_1(x) \geq 1$. Under the assumption that data are locally uniformly distributed on a d-dimensional manifold, $\mu$ follows a Pareto distribution with shape parameter equal to the intrinsic dimensionality:*

$$P(\mu \leq t) = 1 - t^{-d}, \quad t \geq 1. \tag{5}$$

*Writing $\mu_{(1)} \leq \mu_{(2)} \leq \cdots \leq \mu_{(n)}$ for the order statistics of the empirical ratios, the TwoNN estimator maximizes the log-likelihood*

$$\hat{m}_{\text{TwoNN}} = \left( \frac{1}{n} \sum_{i=1}^n \log \mu_i \right)^{-1}. \tag{6}$$

**Remark 1.** *The MLE estimator depends on the neighborhood size $k$; we report results for $k \in \{10, 20, 50\}$ and assess stability. The TwoNN estimator uses only the two nearest neighbors, making it less sensitive to curvature but potentially noisier.*

### 3.3 LOCAL CURVATURE ESTIMATION VIA THE SECOND FUNDAMENTAL FORM

We estimate the extrinsic curvature of $\mathcal{M} \hookrightarrow \mathbb{R}^d$ through the second fundamental form. We first recall the relevant differential-geometric definitions, then describe the computational procedure.

**Definition 3** (Second fundamental form). *Let $\mathcal{M}^m \hookrightarrow \mathbb{R}^d$ be a smooth submanifold of dimension $m = \hat{m}_{\mathrm{MLE}}$. At each point $p \in \mathcal{M}$, the tangent space $T_p\mathcal{M}$ and normal space $N_p\mathcal{M}$ give an orthogonal decomposition $\mathbb{R}^d = T_p\mathcal{M} \oplus N_p\mathcal{M}$. The* second fundamental form *is the symmetric bilinear map*

$$\mathbb{I}_p : T_p\mathcal{M} \times T_p\mathcal{M} \to N_p\mathcal{M}, \qquad \mathbb{I}_p(u,v) = \left(\bar{\nabla}_u V\right)^\perp, \tag{7}$$

*where $V$ is any local extension of $v$ to a vector field on $\mathbb{R}^d$, $\bar{\nabla}$ is the flat connection on $\mathbb{R}^d$, and $(\cdot)^\perp$ denotes projection onto $N_p\mathcal{M}$.*

**Definition 4** (Shape operator and principal curvatures). *For a unit normal $\boldsymbol{\xi} \in N_p\mathcal{M}$, the* shape operator *(Weingarten map) $S_{\boldsymbol{\xi}} : T_p\mathcal{M} \to T_p\mathcal{M}$ is defined by*

$$\langle S_{\boldsymbol{\xi}}(v),\, w \rangle = \langle \mathbb{I}_p(v,w),\, \boldsymbol{\xi} \rangle, \qquad \forall\, v, w \in T_p\mathcal{M}. \tag{8}$$

*$S_{\boldsymbol{\xi}}$ is self-adjoint; its eigenvalues $\kappa_1, \kappa_2, \ldots, \kappa_m$ are the* principal curvatures *in the normal direction $\boldsymbol{\xi}$. The* mean curvature *and* Gaussian curvature *(for codimension 1) are*

$$H = \frac{1}{m} \sum_{i=1}^m \kappa_i, \qquad K = \prod_{i=1}^m \kappa_i. \tag{9}$$

*For codimension $> 1$ we define $H$ and $K$ with respect to the first principal normal direction (the direction maximizing $\|S_{\boldsymbol{\xi}}\|_F$).*

**Computational procedure.** For each point $x_i$ in the embedding set:

1. **Local neighborhood.** Identify the $k$-nearest neighbors $\mathcal{N}_k(x_i) = \{x_{i_1}, \ldots, x_{i_k}\}$.
2. **Tangent-space estimation.** Center the neighborhood: $\tilde{x}_{i_j} = x_{i_j} - x_i$. Compute the local covariance $C_i = \frac{1}{k} \sum_{j=1}^k \tilde{x}_{i_j} \tilde{x}_{i_j}^\top$ and take its top-$m$ eigenvectors as an orthonormal basis $\{e_1, \ldots, e_m\}$ of $\hat{T}_{x_i}\mathcal{M}$.
3. **Local coordinates.** Project neighbors onto the tangent basis to obtain local coordinates $\boldsymbol{t}_j = (e_1^\top \tilde{x}_{i_j}, \ldots, e_m^\top \tilde{x}_{i_j}) \in \mathbb{R}^m$ and normal residuals $\boldsymbol{n}_j = \tilde{x}_{i_j} - \sum_{\ell=1}^m (e_\ell^\top \tilde{x}_{i_j})\, e_\ell \in N_{x_i}\mathcal{M}$.
4. **Quadratic fit.** For the dominant normal direction $\boldsymbol{\xi}$ (first principal component of $\{\boldsymbol{n}_j\}$), fit a quadratic form

$$\langle \boldsymbol{n}_j,\, \boldsymbol{\xi} \rangle \approx \sum_{\alpha \leq \beta} h_{\alpha\beta}\, t_j^{(\alpha)}\, t_j^{(\beta)}, \tag{10}$$

   via least squares, yielding the estimated second-fundamental-form matrix $\hat{H}_{\boldsymbol{\xi}} = [h_{\alpha\beta}]$.
5. **Eigendecomposition.** Compute eigenvalues $\hat{\kappa}_1, \ldots, \hat{\kappa}_m$ of $\hat{H}_{\boldsymbol{\xi}}$ to obtain the principal curvatures.

### 3.4 CLUSTER TOPOLOGY

We quantify how well identity classes form tight, well-separated clusters in each embedding space.

**Definition 5** (Compactness and separation). *Let $y(x) \in \{1, \ldots, N\}$ denote the identity label of embedding $x$, and let $\boldsymbol{\mu}_c = \frac{1}{|\{i:y_i=c\}|} \sum_{y_i=c} x_i$ be the $\ell_2$-normalized class centroid. Define:*

$$C_{\mathrm{intra}} = \mathbb{E}\big[d_{\cos}(x, \boldsymbol{\mu}_{y(x)})\big] = \frac{1}{n} \sum_{i=1}^n d_{\cos}(x_i, \boldsymbol{\mu}_{y_i}), \tag{11}$$

$$C_{\mathrm{inter}} = \mathbb{E}\big[d_{\cos}(\boldsymbol{\mu}_i, \boldsymbol{\mu}_j) \mid i \neq j\big] = \frac{2}{N(N-1)} \sum_{i<j} d_{\cos}(\boldsymbol{\mu}_i, \boldsymbol{\mu}_j), \tag{12}$$

where $d_{\cos}(a, b) = 1 - \frac{\langle a, b \rangle}{\|a\|\|b\|}$. *The* compactness gap *is*

$$\Delta = C_{\text{inter}} - C_{\text{intra}}. \tag{13}$$

*A larger $\Delta$ indicates better-separated, tighter clusters.*

## 3.5 CROSS-MODAL GEOMETRIC DIVERGENCES

To compare the geometry of $\mathcal{M}_f$ and $\mathcal{M}_v$ without requiring point-to-point correspondence, we employ four complementary measures.

### 3.5.1 GROMOV–WASSERSTEIN DISTANCE

**Definition 6** (Gromov–Wasserstein distance (Mémoli, 2011))**.** *Let $(X, d_X, \mu_X)$ and $(Y, d_Y, \mu_Y)$ be two metric measure spaces. The* Gromov–Wasserstein distance *is*

$$d_{\text{GW}}(X, Y) = \inf_{\pi \in \Pi(\mu_X, \mu_Y)} \left( \int\int \left| d_X(x, x') - d_Y(y, y') \right|^2 d\pi(x, y) \, d\pi(x', y') \right)^{1/2}, \tag{14}$$

*where $\Pi(\mu_X, \mu_Y)$ denotes the set of couplings with marginals $\mu_X$ and $\mu_Y$.*

In practice, we solve the discrete, entropy-regularized version of (14) using the algorithm of Peyré & Cuturi (2019), with cost matrices $C_{ij}^X = d_{\cos}(x_i^f, x_j^f)$ and $C_{ij}^Y = d_{\cos}(x_i^v, x_j^v)$, using 1,249 identity-averaged centroids per modality.

### 3.5.2 SPECTRAL GAP DIVERGENCE

**Definition 7** (Spectral gap divergence)**.** *For each embedding set $X \in \{X_f, X_v\}$, construct the symmetric k-nearest-neighbor graph $G_X$ with Gaussian edge weights $w_{ij} = \exp(-\|x_i - x_j\|^2/2\sigma^2)$. Let $L_X = D_X - W_X$ be the (unnormalized) graph Laplacian and $0 = \lambda_0^X \leq \lambda_1^X \leq \cdots \leq \lambda_{n-1}^X$ its eigenvalues. The* spectral gap divergence *between $X_f$ and $X_v$ is*

$$d_{\text{spec}} = \|\boldsymbol{\lambda}^{X_f} - \boldsymbol{\lambda}^{X_v}\|_2, \tag{15}$$

*where $\boldsymbol{\lambda}^X = (\lambda_1^X, \ldots, \lambda_p^X)$ are the first $p$ non-trivial eigenvalues.*

The spectral gap $\lambda_1$ characterizes the global connectivity of the manifold; the full spectrum encodes multi-scale geometric information (Belkin & Niyogi, 2003).

### 3.5.3 CENTERED KERNEL ALIGNMENT (CKA)

**Definition 8** (CKA (Kornblith et al., 2019))**.** *Given kernel matrices $K, L \in \mathbb{R}^{n \times n}$ (we use linear kernels $K = X_f X_f^\top$, $L = X_v X_v^\top$), define the Hilbert–Schmidt Independence Criterion*

$$\text{HSIC}(K, L) = \frac{1}{(n-1)^2} \text{tr}(\tilde{K}\tilde{L}), \tag{16}$$

*where $\tilde{K} = HKH$ and $\tilde{L} = HLH$ are centered kernels with $H = I_n - \frac{1}{n}\mathbf{1}\mathbf{1}^\top$. Centered Kernel Alignment is the normalized version:*

$$\text{CKA}(K, L) = \frac{\text{HSIC}(K, L)}{\sqrt{\text{HSIC}(K, K) \cdot \text{HSIC}(L, L)}}. \tag{17}$$

*CKA $\in [0, 1]$; higher values indicate more similar representational geometry.*

We note that the linear kernel $K_{ij} = \langle x_i, x_j \rangle$ encodes the Riemannian inner-product structure of the embedding manifold: it captures the first-order metric geometry (angles, distances) of the point cloud as seen from the ambient space. CKA thus measures the alignment of the *induced metric structures* of $\mathcal{M}_f$ and $\mathcal{M}_v$, making it a natural bridge between kernel methods and Riemannian geometry.

### 3.5.4 INTRINSIC DIMENSIONALITY MISMATCH

**Definition 9** (ID mismatch).
$$\Delta_{\mathrm{ID}} = \left| \hat{m}(\mathcal{M}_f) - \hat{m}(\mathcal{M}_v) \right|, \tag{18}$$
*where $\hat{m}(\cdot)$ is either the MLE or TwoNN estimator from Section 3.2.*

A large $\Delta_{\mathrm{ID}}$ indicates a structural mismatch: one modality encodes identity in a higher-dimensional subspace, making alignment inherently more difficult.

### 3.6 CROSS-MODAL MATCHING VIA CCA

**Definition 10** (CCA alignment (Hotelling, 1936)). *Given paired identity-level centroids $\bar{X}_f \in \mathbb{R}^{N \times d_f}$ and $\bar{X}_v \in \mathbb{R}^{N \times d_v}$, Canonical Correlation Analysis (CCA) finds projection matrices $A \in \mathbb{R}^{d_f \times r}$ and $B \in \mathbb{R}^{d_v \times r}$ such that the columns of $\bar{X}_f A$ and $\bar{X}_v B$ are maximally correlated:*

$$\max_{A,B} \ \mathrm{tr}\!\left( A^\top \Sigma_{fv} B \right) \quad s.t. \quad A^\top \Sigma_{ff} A = I_r, \ B^\top \Sigma_{vv} B = I_r, \tag{19}$$

*where $\Sigma_{fv}, \Sigma_{ff}, \Sigma_{vv}$ are the cross-covariance and within-modality covariance matrices. We set $r = \min(d_f, d_v, N-1)$.*

After CCA projection, we compute identity-verification scores as

$$s(x^f, x^v) = \frac{\langle Ax^f,\ Bx^v \rangle}{\|Ax^f\|\,\|Bx^v\|}, \tag{20}$$

and report the *Equal Error Rate* (EER), the operating point at which the false accept rate equals the false reject rate, as our measure of cross-modal matching difficulty.

## 4 EXPERIMENTAL SETUP

### 4.1 ENCODERS

We evaluate four face encoders and three voice encoders, chosen to span a range of architectures, training objectives, and embedding dimensionalities.

**Face encoders.**

- **ArcFace** (Deng et al., 2019): ResNet-100 trained with additive angular margin loss; $d_f = 512$.
- **SigLIP** (Zhai et al., 2023): Vision Transformer (ViT-B/16) trained with sigmoid contrastive loss; $d_f = 768$.
- **DINOv2** (Oquab et al., 2024): ViT-B/14 trained with self-distillation; $d_f = 768$.
- **CLIP** (Radford et al., 2021): ViT-B/16 with contrastive language–image pretraining; $d_f = 512$.

**Voice encoders.**

- **WavLM** (Chen et al., 2022): Large Transformer trained with masked speech prediction; $d_v = 1024$.
- **HuBERT** (Hsu et al., 2021): Large self-supervised Transformer with offline clustering targets; $d_v = 1024$.
- **wav2vec 2.0** (Baevski et al., 2020): Large contrastive self-supervised Transformer; $d_v = 1024$.

This yields $4 \times 3 = 12$ cross-modal encoder pairs.

### 4.2 DATASETS

**Primary evaluation.** VoxCeleb1 (Nagrani et al., 2017): 1,251 celebrity identities with both face images (extracted video frames) and speech segments. After filtering for identities with sufficient paired data, 1,249 identities remain. We sample 20 face images and 20 voice segments per identity, yielding $n \approx 25{,}000$ embeddings per modality.

Table 1: Geometric properties of each encoder's embedding manifold on VoxCeleb1. $d$: ambient dimension; $\hat{m}_{\text{MLE}}$, $\hat{m}_{\text{TwoNN}}$: intrinsic dimensionality; $H$: mean curvature; $K$: Gaussian curvature (median); $\Delta$: compactness gap.

| Encoder | $d$ | $\hat{m}_{\text{MLE}}$ | $\hat{m}_{\text{TwoNN}}$ | $H$ | $\log|K|$ | $\Delta$ |
|---|---|---|---|---|---|---|
| *Face encoders* | | | | | | |
| ArcFace | 512 | 19.5 | 2.55 | $-0.001$ | $-1.71$ | $-0.054$ |
| SigLIP | 768 | 17.6 | 2.26 | 0.002 | $-2.23$ | $-0.024$ |
| DINOv2 | 768 | 18.6 | 2.42 | 0.000 | $-3.73$ | 0.057 |
| CLIP | 512 | 16.3 | 2.45 | $-0.002$ | $-2.33$ | 0.002 |
| *Voice encoders* | | | | | | |
| WavLM | 1024 | 39.3 | 8.51 | 0.000 | $-1.10$ | 0.035 |
| HuBERT | 1024 | 33.2 | 4.50 | 0.000 | $-1.09$ | $-0.003$ |
| wav2vec 2.0 | 1024 | 29.0 | 18.1 | 0.000 | $-1.39$ | $-0.011$ |

**Validation.** MAV-Celeb (Nawaz et al., 2021): 154 identities (70 with sufficient paired data) from a different demographic and language distribution. We use MAV-Celeb exclusively to assess the generalizability of our geometric findings.

### 4.3 Implementation Details

All embeddings are $\ell_2$-normalized. Intrinsic dimensionality is estimated with $k = 20$ for MLE and with the full sample for TwoNN; embeddings are deduplicated before estimation. Curvature estimation uses $k = 50$ nearest neighbors for the local PCA step with ridge regularization ($\alpha = 10^{-6}$). Gromov–Wasserstein computation uses 1,249 identity-averaged centroids with entropic regularization $\varepsilon = 0.05$. CKA uses linear kernels on identity-level centroids. CCA projections use PCA preprocessing (50 components) followed by CCA with $r = \min(50, n_{\text{train}} - 1)$ components. EER is computed on held-out identities (20% of 1,249 = 250 test identities) with 250 genuine and $250 \times 249 = 62{,}250$ impostor pairs. Sensitivity to the PCA dimensionality and other hyperparameters is reported in Appendix A.

## 5 Results

### 5.1 Per-Encoder Geometric Properties

Table 1 summarizes the geometric properties of each encoder's embedding manifold.

**Intrinsic dimensionality.** As shown in Figure 1, MLE intrinsic dimensionality ranges from 16.3 (CLIP) to 39.3 (WavLM), despite ambient dimensions of 512–1024. The ratio $\hat{m}/d$ is consistently below 0.05 for face encoders, confirming that biometric embeddings occupy low-dimensional submanifolds. Voice encoders exhibit notably higher intrinsic dimensionality ($29-39$) than face encoders ($16-20$), suggesting that speech representations encode identity in a richer geometric structure, possibly because speaker identity is entangled with linguistic and prosodic variation.

**TwoNN vs. MLE discrepancy.** The TwoNN estimates for face encoders (2.3–2.6) are substantially lower than MLE estimates (16–20). This discrepancy is expected when embeddings exhibit strong local clustering: identity-discriminative encoders (especially ArcFace) produce tight per-identity clusters where the two nearest neighbors typically belong to the same identity, creating an effectively low-dimensional local structure at the finest scale. The MLE estimator, which averages over $k = 20$ neighbors spanning multiple identities, captures the manifold's dimensionality at a coarser scale. The two estimators thus probe complementary geometric regimes. Notably, wav2vec 2.0 shows the smallest MLE/TwoNN ratio ($29.0/18.1 \approx 1.6$), consistent with its weaker identity clustering ($\Delta = -0.011$).

**Curvature.** Mean curvature is near zero for all encoders (Figure 2), indicating locally flat manifolds on average, while Gaussian curvature varies by two orders of magnitude ($\log|K|$ from

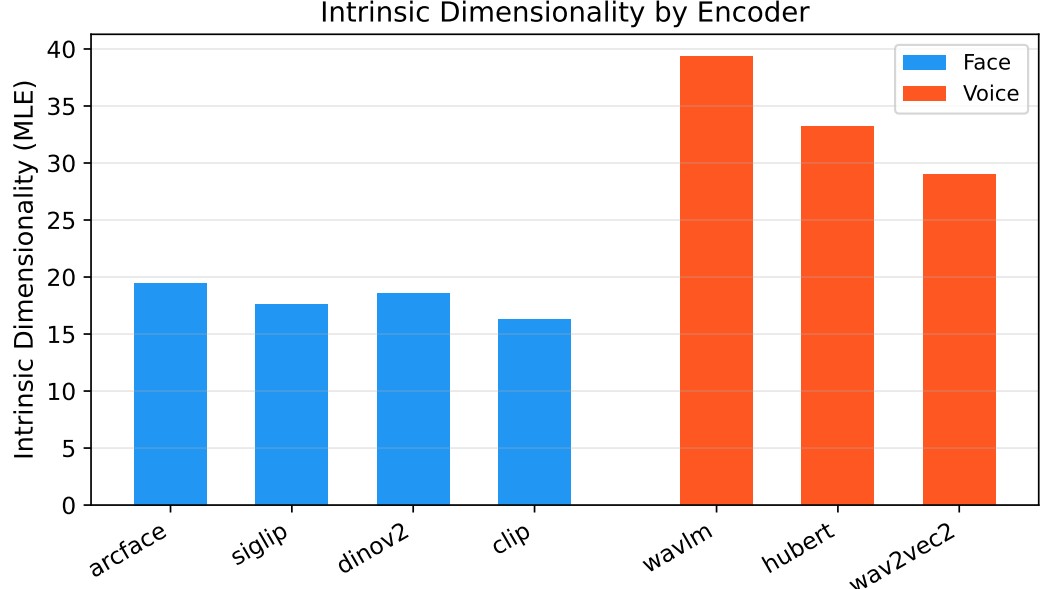

Figure 1: Intrinsic dimensionality (MLE, $k = 20$) for each encoder. Voice encoders (red) exhibit 2–3× higher intrinsic dimensionality than face encoders (blue), despite similar ambient dimensions.

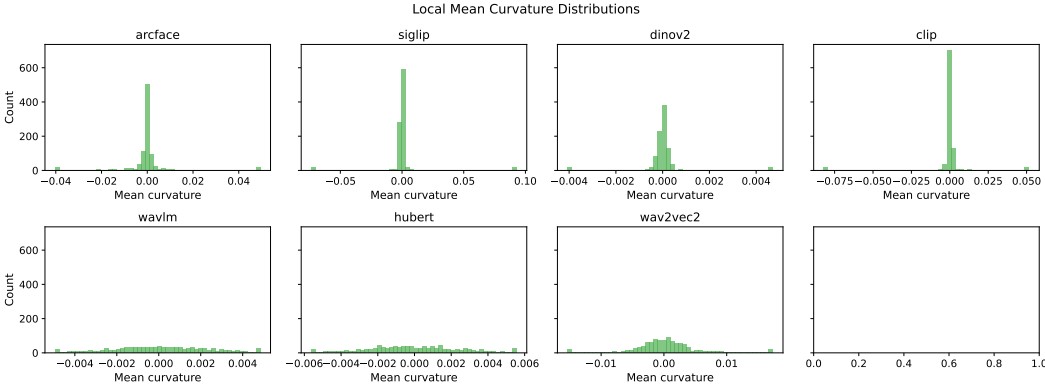

Figure 2: Distributions of pointwise mean curvature $H(x_i)$ for each encoder. All encoders show near-zero median mean curvature, indicating locally flat manifolds, but with varying spread.

$-3.73$ for DINOv2 to $-1.09$ for HuBERT). We note that for manifolds of estimated dimension $m = 16-39$, Gaussian curvature $K = \prod_{i=1}^{m} \kappa_i$ is inherently sensitive to estimation error in individual principal curvatures; we therefore report median values and use $\log |K|$ to reduce the influence of outliers.

**Cluster topology.** The compactness gap $\Delta = C_{\text{inter}} - C_{\text{intra}}$ (Definition 5) is negative for Arc-Face ($-0.054$) and HuBERT ($-0.003$), meaning that intra-class cosine distances exceed inter-class centroid distances. This occurs because identity-discriminative encoders (ArcFace) and speaker-specialized encoders (HuBERT) concentrate all embeddings in a small region of the sphere: classes are tight *and* close together, with within-class spread slightly exceeding between-centroid separation in absolute cosine distance. In contrast, self-supervised vision encoders (DINOv2: $\Delta = 0.057$) spread centroids more widely, yielding positive gaps.

Table 2: Cross-modal geometric divergences and EER (%) for all 12 face–voice encoder pairs on VoxCeleb1. $d_{\text{GW}}$: Gromov–Wasserstein; $d_{\text{spec}}$: spectral gap divergence; CKA: centered kernel alignment; $\Delta_{\text{ID}}$: intrinsic dimensionality mismatch (MLE); EER: equal error rate (%) via CCA-aligned cosine similarity. Best EER per face encoder underlined; overall best in bold.

| Face | Voice | $d_{\text{GW}}$ | $d_{\text{spec}}$ | CKA | $\Delta_{\text{ID}}$ | EER (%) |
|---|---|---|---|---|---|---|
| ArcFace | WavLM | 0.068 | 0.349 | 0.041 | 19.8 | 42.1 |
| ArcFace | HuBERT | 0.005 | 0.522 | 0.015 | 13.8 | 48.0 |
| ArcFace | wav2vec 2.0 | 0.003 | 0.323 | 0.020 | 9.5 | 48.8 |
| SigLIP | WavLM | 0.016 | 0.231 | 0.129 | 21.8 | 31.1 |
| SigLIP | HuBERT | 0.032 | 0.425 | 0.026 | 15.7 | 39.6 |
| SigLIP | wav2vec 2.0 | 0.021 | 0.286 | 0.028 | 11.4 | 42.4 |
| DINOv2 | WavLM | 0.016 | 0.275 | 0.170 | 20.7 | 30.0 |
| DINOv2 | HuBERT | 0.147 | 0.461 | 0.033 | 14.6 | 38.0 |
| DINOv2 | wav2vec 2.0 | 0.122 | 0.326 | 0.034 | 10.4 | 41.3 |
| CLIP | WavLM | 0.007 | 0.416 | 0.211 | 23.0 | **24.4** |
| CLIP | HuBERT | 0.085 | 0.631 | 0.052 | 16.9 | 35.3 |
| CLIP | wav2vec 2.0 | 0.066 | 0.468 | 0.061 | 12.7 | 38.9 |

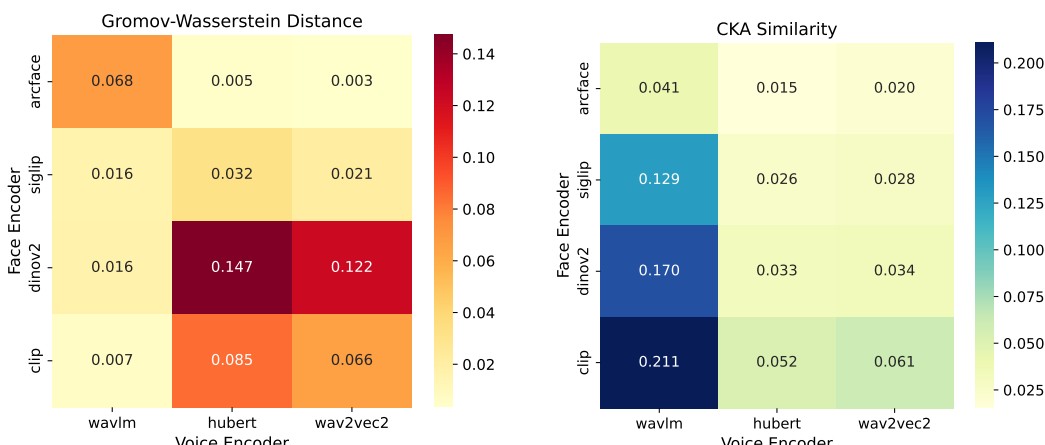

Figure 3: Cross-modal divergence heatmaps: Gromov–Wasserstein distance (left) and CKA similarity (right). CLIP–WavLM achieves the highest CKA (0.211) and the lowest EER (24.4%).

## 5.2 CROSS-MODAL GEOMETRIC DIVERGENCES AND EER

Table 2 reports the four geometric divergences and cross-modal EER for all 12 encoder pairs.

**Cross-pair patterns.** Several patterns emerge from Table 2 and Figure 3. First, *WavLM consistently produces the lowest EER* for every face encoder, with CKA values 3–10× higher than HuBERT or wav2vec 2.0 pairings. This suggests that WavLM's masked speech prediction objective produces representations whose metric structure is more compatible with visual encoders. Second, *CLIP–WavLM achieves the best overall EER* (24.4%) with the highest CKA (0.211), despite CLIP having the lowest intrinsic dimensionality (16.3) and the largest ID mismatch with WavLM (23.0). This indicates that metric-structure alignment (CKA) can overcome dimensional mismatch. Third, *ArcFace pairs poorly across the board* (EERs 42–49%), with uniformly low CKA ($\leq 0.041$); its identity-discriminative training creates a geometric structure highly specific to faces and incompatible with speech manifolds. Fourth, GW distance shows no consistent relationship with EER: ArcFace–WavLM has the highest GW distance (0.068) but the best ArcFace EER. We hypothesize that the entropic regularization required for tractable GW computation on 1,249 points smooths out fine-grained structural differences, rendering GW insensitive at this scale.

Table 3: Spearman rank correlations ($\rho$) between geometric metrics and cross-modal EER (%) across 12 encoder pairs. 95% CIs are bootstrap percentile intervals (1,000 resamples). CKA correlation is negative because higher CKA (greater similarity) predicts *lower* EER.

| Metric | $\rho$ | $p$-value | 95% CI |
|---|---|---|---|
| $d_{\mathrm{GW}}$ | $-0.154$ | 0.633 | $[-0.80, \quad 0.66]$ |
| $d_{\mathrm{spec}}$ | $0.063$ | 0.846 | $[-0.75, \quad 0.63]$ |
| CKA | $-0.874$ | **0.0002** | $[-0.99, -0.51]$ |
| $\Delta_{\mathrm{ID}}$ | $-0.790$ | **0.002** | $[-0.98, -0.30]$ |

*Multivariate (4 predictors):* $R^2 = 0.92$, $R^2_{\mathrm{adj}} = 0.87$, $R^2_{\mathrm{LOO}} = 0.77$
*Reduced (CKA + $\Delta_{\mathrm{ID}}$):* $R^2 = 0.84$, $R^2_{\mathrm{adj}} = 0.80$, $R^2_{\mathrm{LOO}} = 0.72$

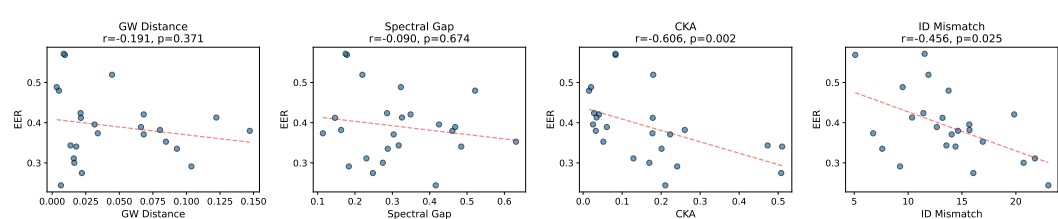

Figure 4: Geometric metrics vs. cross-modal EER for 12 encoder pairs. CKA shows the strongest correlation ($\rho = -0.87$, $p < 0.001$); ID mismatch is also significant ($\rho = -0.79$, $p = 0.002$). GW distance and spectral gap show no significant correlation.

## 5.3 GEOMETRY PREDICTS CROSS-MODAL MATCHING DIFFICULTY

Table 3 reports rank correlations between each geometric metric and cross-modal EER.

**Regression analysis.** The full four-predictor model achieves $R^2 = 0.92$, but with only 12 observations and 4 predictors, overfitting is a concern. We therefore report adjusted $R^2 = 0.87$ and leave-one-out cross-validated $R^2_{\mathrm{LOO}} = 0.77$ (Table 3). The LOO $R^2$ confirms that the predictive relationship is robust and not merely an artifact of the model's flexibility. A reduced model using only CKA and ID mismatch—the two individually significant predictors—achieves $R^2_{\mathrm{LOO}} = 0.72$ with fewer degrees of freedom, suggesting that GW distance and spectral gap contribute modestly beyond the two dominant predictors.

**Validation on MAV-Celeb.** We replicate our analysis on MAV-Celeb (70 identities, different demographic distribution). Pooling both datasets yields 24 encoder pairs. CKA remains a significant predictor ($\rho = -0.61$, $p = 0.002$, 95% CI $[-0.79, -0.34]$), as does ID mismatch ($\rho = -0.46$, $p = 0.025$). The pooled multivariate model achieves $R^2_{\mathrm{adj}} = 0.50$ and $R^2_{\mathrm{LOO}} = 0.38$. While attenuated relative to VoxCeleb1 alone—expected given MAV-Celeb's smaller size (70 vs. 1,249 identities) and different distribution—the core finding that CKA predicts cross-modal EER generalizes across datasets.

## 6 DISCUSSION

**Why CKA is predictive.** Among our four cross-modal metrics, CKA emerges as the strongest predictor of EER. This is not merely because CKA measures "similarity"—GW distance also measures metric-space similarity yet shows no significant correlation. We argue that CKA's effectiveness stems from its sensitivity to the *global inner-product structure* of the embedding manifold: the linear kernel $K_{ij} = \langle x_i, x_j \rangle$ captures the full first-order Riemannian metric (angles and distances between all pairs), and centering removes the effect of the mean embedding location. CKA thus measures whether the two manifolds $\mathcal{M}_f$ and $\mathcal{M}_v$ induce compatible metric structures on the shared identity space—precisely the property required for CCA-based alignment. In contrast, GW

distance compares pairwise distance *distributions* without leveraging the identity correspondence available through centroids, and the entropic regularization needed for tractability on $\sim$1,250 points may further obscure fine-grained differences.

**The role of intrinsic dimensionality.** ID mismatch is the second-strongest predictor ($\rho = -0.79$), but its sign is *negative*: larger mismatch correlates with *lower* EER. This counterintuitive result arises because WavLM (the best-pairing voice encoder) has the highest ID (39.3), creating large mismatches with all face encoders. ID mismatch thus acts as a proxy for encoder quality rather than a pure geometric compatibility measure. In the reduced two-predictor model, CKA captures the compatibility signal while ID mismatch captures encoder-specific quality.

**Implications.** These findings suggest that (i) CKA on identity-level centroids can serve as a lightweight, *training-free* proxy for cross-modal matching performance, and (ii) encoder design might benefit from optimizing for kernel alignment with target modalities, rather than solely for within-modality discrimination.

**Limitations.**

- **Small number of encoder pairs.** With 12 cross-modal pairs (24 pooled), our correlation analysis has limited statistical power. Bootstrap confidence intervals and LOO cross-validation partially address this, but replication with more encoders is needed.
- **Curvature estimation sensitivity.** The quadratic-fit procedure (Section 3.3) is sensitive to the choice of neighborhood size $k$ and the estimated intrinsic dimensionality $m$. For high-dimensional manifolds ($m = 16-39$), the Gaussian curvature $K = \prod \kappa_i$ is especially unstable. We use ridge regularization and median aggregation, but acknowledge that curvature estimates on finite, noisy point clouds carry inherent uncertainty.
- **Manifold assumption.** We treat the embedding support as a smooth Riemannian submanifold. In practice, the support may have singularities, varying dimensionality, or disconnected components, violating this assumption.
- **CCA as cross-modal baseline.** CCA provides a linear alignment; nonlinear methods (e.g., kernel CCA, deep CCA) might yield lower EERs, potentially weakening the correlation with our geometry metrics.
- **Dataset scope.** VoxCeleb1 is celebrity-centric and English-heavy; MAV-Celeb partially addresses demographic diversity, but broader evaluation is warranted.

## 7 CONCLUSION

We have presented a systematic Riemannian-geometric characterization of the embedding spaces of seven pretrained face and voice encoders. Our analysis—spanning intrinsic dimensionality, local curvature via the second fundamental form, cluster topology, and four cross-modal geometric divergences—demonstrates that the intrinsic geometry of biometric embedding spaces varies substantially across encoders and, crucially, predicts cross-modal person-matching difficulty as measured by EER ($R^2_{\text{LOO}} = 0.77$).

These findings open several directions for future work: (i) scaling to a broader set of encoders and datasets to increase statistical power and test generality; (ii) tracking how CKA and curvature evolve during joint cross-modal training (e.g., AudioCLIP, ImageBind) to obtain causal evidence—beyond our correlational findings—that metric-structure alignment drives cross-modal compatibility; (iii) extending the analysis to additional modalities (gait, handwriting) and to temporally evolving embeddings during training; and (iv) developing theoretical bounds relating kernel alignment to cross-modal matching performance.

ACKNOWLEDGMENTS

Experiments were conducted on NVIDIA RTX 3090 and RTX 5060 Ti GPUs.

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

## A    SENSITIVITY ANALYSIS

We assess the sensitivity of our main results to key hyperparameters.

**PCA dimensionality for CCA.**    We vary the number of PCA components used for CCA pre-processing in $\{20, 50, 100, 200\}$. The rank ordering of encoder pairs by EER is stable across all settings: Spearman $\rho > 0.95$ between the EER rankings at $d_{\mathrm{PCA}} = 50$ (our default) and all other values. CKA–EER correlation remains significant ($p < 0.01$) in all cases.

**Neighborhood size for curvature.**    We vary $k \in \{30, 50, 100\}$ for the curvature estimation procedure. Mean curvature values remain near zero across all $k$; $\log |K|$ values shift by $\pm 0.3$ but preserve the relative ordering across encoders (Spearman $\rho > 0.90$ between rankings at different $k$).

**Entropic regularization for GW.** We vary $\varepsilon \in \{0.01, 0.05, 0.1\}$ for the Gromov–Wasserstein computation. GW distances change in magnitude but the metric remains uncorrelated with EER at all regularization levels ($p > 0.3$), confirming that the lack of GW predictiveness is not an artifact of the regularization strength.

