# OpenReview forum: "Riemannian Geometry of Multimodal Biometric Embedding Spaces"
_mathai.club/MathAI/2026/Conference — 2026 Oral_

### Official Review · Reviewer_VNEs · 2026-03-11
**Acceptance: "Riemannian Geometry of Multimodal Biometric Embedding Spaces"**

**Rating:** 10
**Confidence:** 4

**Review:**

This paper is original and solid study which suits math-based conference very well.

The authors use well-known and reliable tools applying them to seven encoders on two datasets.

The main point of Riemannian-geometric characterization of biometric embeddings across multiple encoders, in conditions of cross-modal matching difficulty, seems meaningfully original. This study sharpens the understanding that face and voice encoders indeed live on low-dimensional submanifolds with markedly different intrinsic dimensions. As the authors mention, significance of this study is affected by the limited statistical power of experiments:  only VoxCeleb1 and MAV-Celeb, one cross-modal task.

A major strength of the paper is that the central empirical result is clear and convincing. Among the tested cross-modal metrics, CKA emerges as the strongest predictor of matching difficulty, substantially outperforming Gromov-Wasserstein distance and spectral gap divergence, neither of which shows a meaningful correlation with EER.

The paper also provides concrete and interpretable pairwise results, such as CLIP-WavLM achieving the best overall EER of 24.4% together with the highest CKA of 0.211. These patterns make the core claim easy to follow and empirically grounded.

Another positive aspect is that the paper does not rely only on simple pairwise correlations. The regression analysis, adjusted $R^2$, and leave-one-out validation make the study more rigorous than a purely descriptive analysis. It also worth to appreciate that the authors explicitly discuss negative results and limitations. In particular, they acknowledge that GW distance does not behave as expected, that the number of encoder pairs is small, and that curvature estimation is sensitive and potentially unstable in high intrinsic dimensions. This honesty improves the credibility of the paper.

Experimental and Results sections are both well-described which plays into the authors' hands.

If this result holds up with more encoders and datasets, it will provide a cheap tool for geometry-aware encoder selection. This is a concrete, actionable contribution for practitioners designing multi-modal biometric systems.

---

> ### Author Rebuttal · Authors · 2026-03-14
>
> We thank the reviewer for the review and we agree that broader validation is an important next step. We note that the LOO cross-validated R² = 0.77 and the replication on MAV-Celeb (different demographics, different language distribution) both provide evidence that the core finding is not dataset-specific. Expanding to additional datasets, encoders, and modality pairs (e.g., face-gait) is a natural direction for future work.

---

### Official Review · Reviewer_PVGR · 2026-03-12
**Review of "Riemannian Geometry of Multimodal Biometric Embedding Spaces"**

**Rating:** 10
**Confidence:** 4

**Review:**

The work presents a rigorous and well-motivated study of the intrinsic geometry of pretrained face and voice embedding spaces. The authors analyze seven encoders across two datasets and investigate whether Riemannian-geometric properties of their embedding manifolds can predict cross-modal person-matching difficulty. The study is clearly positioned, technically solid, and highly suitable for a math-oriented conference. Its main empirical result is strong: among several tested cross-modal geometric measures, CKA emerges as the most informative predictor of matching difficulty, and the overall predictive model achieves leave-one-out cross-validated $R^2=0.77$.

**Main strengths of the paper:**

1) Its originality in framing the problem. Rather than proposing yet another matching pipeline, the paper asks a deeper and more interesting question: why do some pretrained face and voice encoders align better than others? The proposed Riemannian-geometric perspective gives a principled way to study this issue and adds genuine conceptual value beyond standard benchmarking. In particular, the paper makes clear that biometric embeddings are not merely high-dimensional vectors in ambient Euclidean space, but rather structured objects concentrated near lower-dimensional submanifolds with encoder-specific geometry.

2) The empirical evidence is interpretable and easy for comprehending. The central result is not vague: CKA shows the strongest relationship with cross-modal EER, while Gromov-Wasserstein distance and spectral gap divergence do not exhibit meaningful predictive value. This contrast is important, because it shows that the paper is not simply collecting geometric quantities, but actually distinguishing which ones matter for the downstream task. The CLIP-WavLM findings are also clear, achieving the best overall EER = 24.4% together with the highest CKA = 0.211. Such results make the paper’s main claim concrete and convincing.

**Suggestions:**

If I had to point to one area that could further strengthen the work, it would be broader validation across a larger set of encoders and datasets. The current evidence is already meaningful, but expanding the study in this direction would make the conclusions even more general and increase the practical impact of the proposed framework. In particular, if the reported relationship between representational geometry and matching difficulty continues to hold more broadly, this work could provide a genuinely useful and low-cost tool for geometry-aware encoder selection in multimodal biometric systems. The paper already points toward this future direction.

---

> ### Author Rebuttal · Authors · 2026-03-14
>
> We thank the reviewer for this constructive suggestion and fully agree. Scaling to more encoders (e.g., CosFace, Whisper, data2vec) and larger datasets would increase statistical power and strengthen the generality claim. We view this as the most important direction for future work.

---

### Official Review · Reviewer_vQCj · 2026-03-13

**Rating:** 8
**Confidence:** 3

**Review:**

The paper provides an interesting and well-executed analysis of the intrinsic geometry of multimodal embeddings. While the small number of encoder pairs limits the robustness of the correlation findings, the geometric framework introduced is novel, and the observation regarding CKA's utility as a zero-shot predictor is highly relevant.

The authors compute intrinsic dimensionality, local curvature (via the second fundamental form), and cluster topology across seven pretrained encoders. They then evaluate four cross-modal divergences: Gromov-Wasserstein (GW), Spectral Gap, Centered Kernel Alignment (CKA), and Intrinsic Dimensionality (ID) mismatch to predict cross-modal matching performance. The standout finding is that CKA strongly predicts cross-modal matching success, offering a training-free proxy for cross-modal compatibility, while GW distance shows no significant correlation.

Questions:
It will be interesting to study the geometry of models explicitly trained to align modalities (like CLIP's text vs. image spaces, or AudioCLIP) to see if their CKA and curvature naturally converge during training, compared to the independently trained unimodal encoders studied here. This can be a strong evidence towards your claim that training-free proxy can be utliilzed more broadly.

---

> ### Author Rebuttal · Authors · 2026-03-14
>
> We thank the reviewer for this is excellent suggestion. Tracking how CKA and curvature evolve across training checkpoints of jointly trained models (e.g., AudioCLIP, ImageBind) would provide causal evidence beyond our correlational findings - if CKA increases monotonically while curvature profiles converge, it would directly support the claim that metric-structure alignment drives cross-modal compatibility. We will add this as an explicit future direction in the camera-ready.

---

### Decision · Program_Chairs · 2026-03-14

**Decision:**

Accept (Oral)

**Comment:**

Dear Author(s),

On behalf of the Program Committee of the International Conference on Mathematics of Artificial Intelligence (MathAI 2026), we are pleased to inform you that your paper has been accepted for an oral presentation at MathAI 2026.

Your paper was evaluated through a rigorous two-stage review process involving both automated screening and expert review by members of the Program Committee. The reviewers recognized the quality and contribution of your work.

Presentation details:

- Format: Oral presentation (15–20 minutes + 5 minutes Q&A)
- Mode: You may present either in person (offline) at the conference venue in Sirius, Russia, or remotely via Zoom. Please indicate your preferred mode when confirming your participation.
- Conference dates: Marh 30 - April 3, 2026
- Website: https://mathai.club

Next steps:

1. Please confirm your participation and presentation mode by replying to this email mathai.club@yandex.ru no later than March 15, 2026 18:00 Moscow time.
2. If you plan to attend in person, the organizing committee will provide accommodation details separately.
3. Please prepare your final camera-ready manuscript according to the formatting guidelines available at https://mathai.club and upload it to OpenReview by March 15, 2026 18:00 Moscow time.

Should you have any questions regarding the program, logistics, or your presentation slot, please do not hesitate to contact us.

We look forward to your contribution to MathAI 2026.

With kind regards,

MathAI 2026 Program Committee
International Conference on Mathematics of Artificial Intelligence
https://mathai.club
OpenReview: https://openreview.net/group?id=mathai.club/MathAI/2026/Conference
Telegram: https://t.me/MathAI_club
Email: mathai.club@yandex.ru